# Histidine-Rich Defensins from the *Solanaceae* and *Brasicaceae* Are Antifungal and Metal Binding Proteins

**DOI:** 10.3390/jof6030145

**Published:** 2020-08-24

**Authors:** Mark R. Bleackley, Shaily Vasa, Peta J. Harvey, Thomas M. A. Shafee, Bomai K. Kerenga, Tatiana P. Soares da Costa, David J. Craik, Rohan G. T. Lowe, Marilyn A. Anderson

**Affiliations:** 1Department of Biochemistry and Genetics, La Trobe Institute for Molecular Science, La Trobe University, Melbourne, VIC 3086, Australia; m.bleackley@latrobe.edu.au (M.R.B.); S.Vasa@latrobe.edu.au (S.V.); T.Shafee@latrobe.edu.au (T.M.A.S.); bkkerenga@gmail.com (B.K.K.); T.SoaresdaCosta@latrobe.edu.au (T.P.S.d.C.); R.Lowe@latrobe.edu.au (R.G.T.L.); 2Institute for Molecular Bioscience, The University of Queensland, Brisbane, QID 4072, Australia; peta.harvey@imb.uq.edu.au (P.J.H.); d.craik@imb.uq.edu.au (D.J.C.)

**Keywords:** plant defensin, antifungal, metal binding, histidine

## Abstract

Plant defensins are best known for their antifungal activity and contribution to the plant immune system. The defining feature of plant defensins is their three-dimensional structure known as the cysteine stabilized alpha-beta motif. This protein fold is remarkably tolerant to sequence variation with only the eight cysteines that contribute to the stabilizing disulfide bonds absolutely conserved across the family. Mature defensins are typically 46–50 amino acids in length and are enriched in lysine and/or arginine residues. Examination of a database of approximately 1200 defensin sequences revealed a subset of defensin sequences that were extended in length and were enriched in histidine residues leading to their classification as histidine-rich defensins (HRDs). Using these initial HRD sequences as a query, a search of the available sequence databases identified over 750 HRDs in solanaceous plants and 20 in brassicas. Histidine residues are known to contribute to metal binding functions in proteins leading to the hypothesis that HRDs would have metal binding properties. A selection of the HRD sequences were recombinantly expressed and purified and their antifungal and metal binding activity was characterized. Of the four HRDs that were successfully expressed all displayed some level of metal binding and two of four had antifungal activity. Structural characterization of the other HRDs identified a novel pattern of disulfide linkages in one of the HRDs that is predicted to also occur in HRDs with similar cysteine spacing. Metal binding by HRDs represents a specialization of the plant defensin fold outside of antifungal activity.

## 1. Introduction

Plant defensins are a remarkable family of proteins. They are defined by a conserved three-dimensional structure consisting of three beta strands and a single alpha helix stabilized by four disulfide bonds forming a fold known as the cysteine-stabilized alpha-beta (CSαβ) motif [1]. The amino acid sequence requirements to form the CSαβ motif appear to be limited to the eight cysteines that participate in the disulfide bonds, although the spacing between these cysteines can vary, with the only other residues that are more than 80% conserved being two glycines. The utility of the plant defensin fold in biology is demonstrated by the presence of defensin sequences in the genomes of all plant species; in some cases, there are hundreds of defensin sequences in a single plant genome. Many of the plant defensins that have been characterized functionally have antifungal activity, thus plant defensins have been classified as innate immunity peptides [2]. However, there are instances of plant defensins with other biological functions, including roles in plant reproduction, signaling, metal tolerance, antibacterial activity, insecticidal activity, and inhibitors of hydrolytic enzymes [3,4].

We assembled a database of approximately 1200 plant defensin and related amino acid sequences and analyzed their sequence characteristics [5,6]. Plant defensins have highly variable sequences, but the mature defensin domain is typically around 50 amino acids in length, cationic, and has a negative hydrophobicity index. However, there are outliers in length, net charge, and amino acid bias. We were particularly interested in defensins that were enriched in histidine residues. Histidine is a unique amino acid due to its imidazole sidechain, which changes charge at physiologically relevant pHs. Histidine residues are also ideally suited to metal binding because they provide two N-donors and a six membered chelate ring for coordination. This property of the histidine side chain means that many proteins that function in metal homeostasis are enriched in histidine.

Both defensins and histidine rich proteins contain examples with annotated functions in plant metal tolerance. Defensins from *Arabidopsis thaliana* [7] and rice [8,9] function in cadmium tolerance and a defensin from *Arabidopsis halleri* [10] has a role in zinc tolerance. However, none of these defensins contain more than two histidine residues. Citrus dehydrins are histidine-rich metal-binding proteins that are expressed in response to osmotic stress and are proposed to reduce metal toxicity in citrus plants under water stressed conditions [11]. A role for the histidine rich defensins in plant defence cannot be discounted as a cysteine/histidine rich DC-1 domain protein from *Capsicum annum* has a role in protecting plants against microbial pathogens [12]. In another example, the peptides shepherin I and II from Shepherds purse, *Capsella bursa-pastoris* are enriched in glycine and histidine residues and are active against Gram-negative bacteria and fungi. These peptides of 28 and 38 amino acids respectively are derived from a larger precursor and have a random coil structure [13]. This overlap between antimicrobial and metal binding activities is also seen in histatin-5, a histidine rich antimicrobial peptide in human saliva, which binds to Zn and Cu [14,15,16].

To determine the potential functions of HRDs we extended our search of available sequence databases using the *N. benthamiana* HRDs as a query sequence. Over 750 distantly related sequences were identified in Solanaceous plants and 20 were identified in brassicas. We selected six HRD sequences, two of the initial *N. benthamiana* sequences, two from other solanaceous plants and two from brassicas, for recombinant expression and assessment of their antimicrobial and metal binding properties. Of the six sequences selected four were successfully expressed and purified. All four displayed some metal binding activity supporting a potential role in metal sequestration. Two of the HRDs also had antifungal activity, demonstrating an overlap between metal binding and antifungal functions of histidine rich plant defensins.

## 2. Materials and Methods

### 2.1. Identification of Additional HRDs

The initial identification of HRDs as part of a wider defensin database and initial principal component analysis is described in [6]. This dataset contains plant, fungal, and invertebrate defensins and homologous sequences. This dataset was extended via a more focused search for additional HRDs. The *N. benthamiana* genome was also probed using the NbD3 sequence (Niben101Scf01052g01004.1) to identify additional HRDs. BLAST searches of the Joint Genome Institute phytozome database (https://phytozome.jgi.doe.gov) were performed using NbD3 and NbD19 (Niben101Scf01052g01005.1) sequences from *N. benthamiana* and AtD90 (AT3G05727) and AtD212 (AT3G05730) sequences from *A. thaliana,* retrieving 14 HRDs from Solanaceae plants and 11 from the Brassicaceae. Signal peptides were removed before the two groups were separately aligned and then used as queries in a Hidden Markov Model using the EMBL-EBI HMMSearch tool (https://www.ebi.ac.uk/Tools/hmmer/search/hmmsearch) restricted to reference proteomes database and viridiaeplantae with the sequence bias filter turned off. This search used the BLOSUM60 substitution matrix. The HMMER options used were: --E 1 --domE 1 --incE 0.01 --incdomE 0.03 --nobias --seqdb uniprotrefprot --seqdb_ranges 134108309..139436241. The resulting sequences were added to the HRD dataset (239 from Solanaceae and 18 sequences from Brassicaceae). The combined sequence set was then subjected to redundancy reduction to remove any sequences with ≥99% identity, yielding a final set of 1901 sequences (1628 from the previous dataset in [6], 273 additional sequences from the additional searches for this study).

### 2.2. Sequence Analysis

The sequence set was aligned using Clustal Omega (https://www.ebi.ac.uk/Tools/msa/clustalo/), [17], constrained by CysBar to ensure alignment of homologous cysteines [5]. The multiple sequence alignment was used to generate a projected sequence space via the same numericization and multidimensional scaling process as published previously for the superfamily [6]. Phylogenies for defensins are notoriously difficult to construct due to their low sequence conservation. The closest 30 sequences to NbD2, AtD212, and SlD26 within the sequence space were gathered and maximum likelihood phylogenies attempted. The substitution model was identified and 1000 bootstrap phylogenies were generated with the Phangorn and Ape packages in R [18,19].

### 2.3. Confirmation of Expression of HRDs in Planta

To confirm that the HRDs we had identified in *N. benthamiana* and *A. thaliana* were expressed *in planta* and were not pseudo genes we searched the *N. benthamiana* Gene Expression Atlas (https://sefapps02.qut.edu.au/atlas/tREX6.php) and the Klepikova Arabidopsis Atlas eFP Browser [20] (http://bar.utoronto.ca/) via The Arabidopsis Information Resource (https://www.arabidopsis.org) to identify the temporal and spacial expression of HRDs.

### 2.4. Cloning, Expression, and Purification of HRDs

DNA sequences corresponding to the mature defensin domain of six HRDs, NbD2 (*Nicotiana benthamiana*, Niben101Scf03038g07007.1), NbD3 (*N. benthamiana* Niben101Scf01052g01004.1), AtD90 (*Arabidopsis thaliana,* AT3G05727), AtD212 (*A. thaliana*, AT3G5730), CrD26 (*Capsella rubella,* Carubv10014946), and SlD26 (*Solanum lycopersicum*, Solyc07g009230) were ordered from Genscript (Piscataway, NJ, USA) that were codon optimized for expression in yeast. The coding sequences of the mature peptides, lacking the predicted signal peptide, were amplified by PCR using primers and cloned into the pPink-alpha-HC vector for expression in the *Pichia pastoris* pPINK system as described in [21] using the primers listed in Appendix A. This expression system adds an additional Alanine residue to the N-terminus. Defensins were initially purified using the standard procedure for plant defensins as described in [22]. This procedure was subsequently optimized for each HRD by determining the best pH for binding to the SP sepharose resins by screening different buffers for equilibration/wash and elution. After induction of expression, the culture supernatant was adjusted to pH 7, 6, 4, or 3 prior to binding to the column by addition of 50 mL of 1 M bis-Tris pH 7, 1 M sodium phosphate buffer pH 6, 1 M sodium acetate buffer pH 4, or 1 M sodium citrate buffer pH 3, respectively to 1 L of supernatant. The equilibration/wash buffers tested were 50 mM potassium phosphate pH 6, 50 mM acetate pH 4, and 20 mM citrate pH 3. Elutions were performed using the same buffer as the equilibration/wash buffer containing 0.5 M NaCl. Protein yields were determined by bicinchoninic acid (BCA) assay (Pierce). Protein quality was assessed by RP-HPLC using an Agilent 1200 system and the expected mass was verified using a Bruker Daltonics Ultraflex III MALDI-TOF/TOF Mass Spectrometer and SDS-PAGE.

### 2.5. Antifungal Activity Assays

The antifungal activity of the HRDs was assessed against the agricultural pathogen *Fusarium graminearum* isolate PH-1 and the human pathogen *Candida albicans* strain ATCC90028. Antifungal assays and fungal culture were performed as described in [23] and [21] for *F. graminearum* and *C. albicans,* respectively.

### 2.6. Metal-induced Precipitation

HRDS were prepared at 80 µM in sterile distilled water and metal ions at 20 mM in 20 mM bis-Tris, 150 mM NaCl, pH 7.4. The HRD and metal ion solution were mixed at a 1:1 ratio (10 µL of each) in 0.2 mL tubes by gentle pipetting prior to incubation at room temperature for 25 min. Metal-induced precipitation was assessed by the eye. Metal-induced precipitation experiments were performed using NiCl_2_ and ZnCl_2_ along with NaCl as control. To confirm that the metal ions were the cause of the precipitation, EDTA was added to each precipitate and redissolution of the precipitate was monitored. To determine whether the HRD protein was present in the precipitate, the tubes were centrifuged at low speed for 30 s and then proteins in the pellet and supernatant were separated by SDS-PAGE and visualized using RapidStain (Millipore, Burlington, MA, USA).

### 2.7. Microscale Thermophoresis

Microscale thermophoresis (MST) was performed with the Monolith NT.LabelFree instrument (NanoTemper Technologies, Munich, Germany) [24,25]. To remove residual metal ions, purified HRDs were incubated with 5 mM EDTA for 30 min before buffer exchange into Chelex (Biorad, Gladesville, NSW, Australia) treated Milli-Q water using a 3 kDa cutoff Amicon spin column (Merck, Darmstadt, Germany). Protein stocks were prepared at 40 µM (SlD26) or 80 µM (AtD90) in the appropriate MST buffer; either 20 mM HEPES, 150 mM NaCl, pH 7.4 for NiCl_2_ assays, or 20 mM bis-Tris, 150 mM NaCl, pH 7.4 for all other metal salts. A fifteen step, two-fold dilution series of each metal salt (and a no metal control) in the respective MST buffer with a top concentration of 20 mM was also prepared. Protein and metal salt solutions were combined in a 1:1 ratio (10 µL of each) in a 0.2 mL tube and mixed by gentle pipetting before incubation for 30 min at room temperature. The samples were then loaded into Monolith NT Standard Treated capillaries (NanoTemper Technologies) via capillary action. Thermophoresis was monitored by intrinsic fluorescence at 25 °C with experiments performed at 20% LED power and 40% MST IR laser power. Data from three independent experiments were analyzed using the signal from Thermophoresis + T-jump employing the NT.Analysis software version 1.5.41 (NanoTemper Technologies).

### 2.8. NMR Spectroscopy and Structural Analysis

NMR spectra of AtD90 and SlD26 were acquired on a Bruker Avance III 600 MHz NMR spectrometer using peptide dissolved in H_2_O/D_2_O (10:1, *v*/*v*) at a concentration of 1 mM and pH 4.0. 1D ^1^H spectra and 2D TOCSY, NOESY, and ^1^H-^15^N HSQC were measured at 298 K, and additional TOCSY spectra at temperatures of 283–308 K were used to identify temperature-dependent amide shifts. Peptide was also dissolved in 100% D_2_O for deuterium exchange experiments and acquisition of ^1^H-^13^C HSQC and ECOSY spectra. Spectra were referenced to an internal standard of 2,2-dimethyl-2-silapentone-5-sulfonate (DSS) at 0 ppm. Spectra were processed using Topspin 3.5 (Bruker) and assigned using Ccpmr Analysis.

Preliminary structures were generated with CYANA 3.97 using distance restraints derived from NOESY spectra (200 ms mixing time), disulfide restraints, and torsion angle restraints generated from TALOS-N [26] and chemical shift assignments. Several chi1 side-chain restraints were added as predicted by ECOSY and NOESY data. CNS [27] then generated a final set of structures using torsion angle dynamics, refinement and energy minimization in explicit solvent. Stereochemical quality of the final structures was assessed using MolProbity [28].

### 2.9. Molecular Modeling

Since the distribution of histidine residues throughout the NbD2 defensin domain differed to those for which structures were solved, a simple homology model was generated using SWISS-MODEL [29]. SPE10 (PDB:3PSM) was chosen as the best-fit structurally characterized template. Note that the exact orientation of the long his-rich loop is low-certainty, as this had to be modeled de novo.

## 3. Results

### 3.1. Identification of Histidine-Rich Defensins

The first three hisitidine-rich defensins we identified were from *N. benthamiana* (NbD1, NbD2, and NbD3) (Appendix A). They were initially recognized as outliers in our database of >1200 defensins based on their extended length relative to other defensins, particularly in loop 5 (Figure 1C) and their relatively high content of histidine residues. Histidine residues are normally rare in defensins (mean 1.5 per sequence) and therefore we defined a new family of histidine-rich defensins (HRDs) using a semi-arbitrary cutoff as those containing six or more His residues in the mature defensin domain (Figure 1A). Using this definition, we went back to the database and identified additional HRDs from *A. thaliana*, AtD90, and AtD212 (Appendix A). Further BLAST searches of plant genomes on the JGI phytozome database using HRD sequences from *N. benthamiana* and *A. thaliana* as queries expanded the list of HRDs to 25, with all the sequences identified in Solanaceous or Brassica plant genomes. Alignments of the 25 HRD sequences were used as a query in a Hidden Markov Model search of plant proteomes on the EMBL-EBI database. This retrieved 239 additional unique sequences from Solanceae and 18 sequences from Brassicaceae.

Querying the *N. benthamiana* Gene Expression Atlas and Klepikova Arabidopsis Atlas eFP Browser confirmed that these genes were transcribed. Of the *A. thaliana* HRD transcripts, AtD90 was expressed in the meristem up until 14 days post-germination, while AtD212 was expressed in the seedling cotyledons and immature leaf blade. For NbD2, gene expression was highest in the seedling and root samples (Figure 1E). The proteins encoded by these transcripts have not been studied in planta. The levels of protein that accumulate and their functions have not been described.

### 3.2. Evolutionary Analyses

Short proteins with highly divergent sequences such as defensins cannot be analyzed with traditional phylogenetics (resulting trees have average bootstraps <20%). Specialized sequence space methods have therefore been developed to overcome these limitations [6,34,35]. When a sequence space of the plant defensin dataset is generated, the majority of HRDs fall into two main clusters (Figure 1B), one containing NbD2 and most of the Solanaceae HRDs, the other containing CrD26, AtD90, and AtD212 and the Brassicaceae HRDs. Isolated HRDs from the Solanaceae, Fabaceae, Poaceae, and Zosteraceae are also distributed outside of these clusters. They include SlD26 from *Solanum lycopersicon* which sits closest to, but distinct from, the Solanaceae HRD cluster. Those HRDs that fall within the two main clusters are mainly histidine-rich in their loop 5 regions, whereas HRD sequences outside of those clusters have histidines distributed throughout their sequences (Figure 1D and Appendix A). The 30 closest sequences to NbD2, AtD90, and SlD26 were used to generate local phylogenies of the most closely related sequences to these, but bootstrap values (mean 40–45%) were too low to yield useful insight.

### 3.3. Expression and Purification of HRDs

To assess the potential biochemical functions of HRDs we selected sequences, NbD2 and SlD26 from the Solanaceae and CrD26, AtD90, and AtD212 from the Brassicaceae, for recombinant expression in *P. pastoris*. Previous transcription studies were checked to confirm the in vivo expression of these genes as an initial filter for biological relevance (Figure 1E). These defensins vary in histidine content, overall length, and isoelectric point (Table 1). Due to the range of predicted pI values for these defensins we modified the standard ion exchange protocol for purification of recombinant defensins from *P. pastoris* supernatants by varying the pH at which the purification was performed. Four buffer systems with different pHs were selected and trialed for purification of HRDs. A summary of these purification trials is presented in Table 2. The buffers selected for large scale purification of each defensin were sodium acetate pH 4 for AtD90, sodium citrate pH 3 for AtD212 and CrD26, and sodium phosphate pH 6 for SlD26. AtD90, AtD212, CrD26, and SlD26 expressed very well with yields of 5–16 mg/L of culture (Table 2). NbD2 expressed poorly and the yield of purified protein was insufficient for further experimentation. The masses of the proteins were confirmed by SDS-PAGE, RP-HPLC, and MALDI-TOF MS.

### 3.4. Antifungal Activity of HRDs

To assess whether HRDs had antifungal activity we tested them in microbroth dilution assays against the human fungal pathogen *C. albicans* and the cereal pathogen *F. graminearum*. SlD26 and AtD90 inhibited the growth of both pathogens but were less active than the well characterized plant defensin NaD1. AtD212 and CrD26 did not have antifungal activity against either species at concentrations up to 50 µg/mL (Table 3).

### 3.5. Metal Binding by HRDs

As histidine residues are able to form coordination complexes with metal ions we assessed whether the HRDs bound to metal ions. Initially this was done using a simple metal-induced precipitation assay where purified HRDs are mixed with metal ions and monitored for the formation of a precipitate. AtD212 and CrD26 both precipitated in the presence of NiCl_2_ and ZnCl_2_. Formation of the AtD90 precipitate was dependent on the concentration of metal ions in the solution. To confirm that the precipitate was protein and not just the metal ions and/or another buffer component the NiCl_2_ samples were centrifuged and the pellet and supernatant were analyzed by SDS-PAGE. Staining the gels with Rapid Stain revealed that the precipitates contained the majority of the protein (Figure 2).

Not all protein-metal interactions lead to protein precipitation. HRDS that did not precipitate in the presence of metal ions, AtD90 and SlD26, were amenable to assessment of metal binding activity using the more precise method of microscale thermophoresis (Table 4). Both AtD90 and SlD26 bound to NiCl_2_ and ZnCl_2_ while only AtD90 bound to MnCl_2_. SlD26 had an approximately ten-fold higher affinity for NiCl_2_ than AtD90, whereas AtD90 had a slightly higher affinity for ZnCl_2_ than SlD26.

### 3.6. Structural Analysis

In order to confirm that the recombinant HRDs had the conserved plant-defensin fold, in the solution NMR structures were solved for SlD26 and AtD90. The 3D solution structures of AtD90 and SlD26 were determined using 552 and 419 distance restraints, respectively, generated from NOESY spectra acquired at 298 K. The first of two prolines (Pro11) in the sequence of AtD90 was assigned as having a trans peptide bond whilst that of Pro14 was determined as cis on the basis of inter-residue NOE correlations. A number of dihedral angle restraints for both peptides were used (see Appendix A) as predicted by TALOS-N along with Chi1 angle restraints based upon ECOSY and NOESY intensities. Disulfide restraints were added based upon homology with other plant defensins that typically have the CSαβ motif represented by the following motif: C1…C2XXXC3…C1′…C2′XC3′ [36]. In the case of SlD26, disulfide restraints were therefore set as Cys15-Cys35, Cys20-Cys40, and Cys24-Cys42 with a fourth disulfide being set between the first and final cysteines (Cys4-Cys46). The sequence of AtD90 suggested that it had a different disulfide pattern to the standard plant defensin. This was confirmed by analysis of preliminary structures calculated with no disulfide restraints, which suggested connectivity for AtD90 of: Cys3-Cys32, Cys15-Cys36, Cys20-Cys45, and Cys24-Cys47 (Figure 3C). This disulfide connectivity also leads to greater C-terminal flexibility than in other plant defensins. Hydrogen bond constraints were added after consideration of preliminary structures, temperature coefficients of amide protons, and deuterium exchange experiments; a total of 8 and 12 hydrogen bond pairs were added for AtD90 and SlD26, respectively. As shown in Appendix A, the final family of structures for both defensins overlay well with good structural and energy statistics. AtD90 and SlD26 each adopt a three turn α-helix tethered to a triple stranded antiparallel β-sheet by disulfide bonds, although the β-strands are longer in AtD90 (Figure 3). The additional disulfide constrains the N- and C- termini of SlD26, whereas it connects the N-terminus of AtD90 to its β2 strand. The assigned chemical shifts of both AtD90 and SlD26 have been deposited in the BMRB (accession codes 30783 and 30784, respectively) and structural coordinates have been deposited in the PDB (7JN6 and 7JNN, respectively).

The sidechain orientation of the AtD90 and SlD26 structures reveal that the majority of histidines are on the same face of the protein (Figure 4B,C). Although NbD2 has a very different sequence, a homology model suggests that the histidines are similarly spatially clustered, but on the opposite side of the structure (Figure 4D).

## 4. Discussion

Histidine-rich plant defensins are a new subclass of plant defensins that in some cases have both antifungal and metal binding activity. All four of the HRDs assessed in this study, AtD90, AtD212, CrD26, and SlD26, bound metal ions but only AtD90 and SlD26 had antifungal activity. HRDs were found exclusively in the *Solanaceae* and *Brassicaceae* families and phylogenetic analysis indicated that they have arisen through two independent evolutionary lineages.

HRDs were initially identified from the genome sequences of *N. benthamiana* and *A. thaliana*. It was advantageous that the initial discovery was in these two well characterized plant species as it permitted the use of published highly detailed gene expression maps to determine where and when the HRDs were transcribed. In both plant species the highest levels of HRD transcript were detected in young plants. Expression levels decreased with plant maturation but transcripts were detected in roots and leaves in both plants, as well as senescent organs of *A. thaliana*. This pattern of expression is consistent with a role for HRDs in protection against either pathogenic attack or metal toxicity during the developmental stages when the plant is most susceptible to disease [37] and metal toxicity [38]. We found no reports on the phenotype of AtD90 or AtD212 mutants of *A. thaliana* in the literature. However, we noted a report that AtD90 gene expression is induced by OXS2 in response to oxidative and salt stress [39]. In addition, AtD212 was differentially regulated in the Arabidopsis response to exposure to selenium, a non-metal with properties similar to arsenic [40]. Together these two reports reveal two instances where HRDs are regulated by environmental stress, including salt and selenium drivers.

Environmental stress may have also contributed to the evolution of the HRDs in *Nicotiana benthamiana*. This plant grows among rocks and cliffs in the metal rich soils of Central and Northern Australia and during its evolution sacrificed defence against viral pathogens for early vigour to ensure survival in the extreme environmental conditions of its natural habitat [41]. This evolutionary trade-off furthered our hypothesis that HRDs had evolved through a similar sacrifice of antifungal activity in favour of metal binding and therefore metal tolerance. This trade off may be reflected in the observations that some of the HRDs had an extension of their loop 5 region and this region was rich in histidine residues. Loop 5 is a key determinant in the antifungal activity of plant defensins [42].

Defensins with slightly more histidine residues than standard defensins occur at random in a wide range of plant families. These HRDs tend to have histidines distributed throughout their sequence and they are scattered throughout the sequence space, indicating a separate evolutionary origin. However, there have been two examples of HRD gene family expansions that form the two clear clusters of HRDs in the sequence space: One in the Solanaceae (e.g., NbD2) and one in the Brassicaceae (e.g., AtD90). Indeed, AtD90 and AtD212 are neighbours in the *A. thaliana* genome, representing a likely paralog duplication event. All HRDs with seven or more His residues sit within these two clusters, and in both cases they show particular histidine enrichment in loop 5, which may indicate a specific selection.

Of the four HRDs that were analyzed in this study, only two, SlD26 and AtD90 had antifungal activity. There was no correlation between the number of histidine residues and antifungal activity as both AtD90 and AtD212 have nine histidine residues. From the small sample size of HRDs in this study there was a link between pI and antifungal activity. Both AtD90 and SlD26 have a pI greater than 6 whereas AtD212 and CrD26 have a pI of less than 6. A positive charge is a defining feature of antimicrobial peptides [43] and is thought to facilitate the initial interaction between the peptide and the microbial cell surface. The lack of positive charge in AtD212 and CrD26 at biological pH is a likely explanation for their lack of antifungal activity.

The metal binding activity of all four HRDs in this study leads to the possibility that they have a function in metal tolerance. Metal binding has been identified in plant defensins before. AhPDF1.1 from *A. halleri* binds to zinc and confers zinc tolerance when expressed in either *Saccharomyces cerevisiae* or *A. thaliana* [10]. However, the mechanism by which AhPDF1.1 confers zinc tolerance is not known. AtPDF2.6 and CAL1 have been linked to cadmium tolerance in *A. thaliana* and rice, respectively through chelation of the toxic metal ion [7,8]. Cadmium binds to these defensins at an approximate 1:1 ratio but no values for metal affinity have been published for either defensin so a comparison of the metal affinity for the HRDs with known metal binders is not possible. Other metal binding peptides or small proteins such as phytochelatins (*K*_d_ < 1 µM) [44] or citrus dehydrins (*K*_d_ 1–27 µM) [11] have higher affinities for metal ions than those we observed for HRDs (*K*_d_s on the order of 100 of µM). However, different metals were tested in the phytochelatin and citrus dehydrin studies. HRDs bind to Ni^2+^, Zn^2+^, and Mn^2+^, which are metals that are both beneficial and toxic to plants, as opposed to Cd^2+^, which is not required by plants and is only present as a toxic environmental contaminant. The lower metal binding affinity for the HRDs may reflect their function in protection against very high levels of essential metals as opposed to low levels of non-essential metals.

The molecular details of how HRDs bind to metal ions is not known. Some metal binding proteins chelate metal ions through their cysteine residues, for example phytochelatins [45] and reduced human beta-defensins [46]. The cysteines of HRDs are all involved in disulfide bonds that define the defensin structure and are not predicted to interact with the metal ions. Rather, the metal binding of the HRDs is proposed to be a function of the histidine residues, similar to citrus dehydrins [11]. While SlD26 hosts these histidines on a standard plant defensin disulfide pattern, the connectivity of AtD90 represents a novel disulfide pattern, adding to the fold’s documented tolerance to changes in disulfide connectivity [47,48]. Based upon the sequences of AtD212 and CrD26, their structures are likely to be more similar to AtD90 than SlD26, albeit with longer inter-cysteine loops. If HRDs form dimers similar to those observed for several antimicrobial defensins [22], the His-rich face of the molecule may drastically affect its metal chelation interactions (Figure 4A).

Plant defensins are traditionally considered to be a component of the pathogen defence system [1]. However, there are often multiple [4], even hundreds of defensin genes in a single genome [49]. It follows that the function of some of these paralogs may be re-specialized, lose antipathogen activity, or become pseudogenes depending on selective pressures on the species. We have now identified a rare specialization based on histidine enrichment, which represents a novel route to metal binding activity in the defensin superfamily.

## Figures and Tables

**Figure 1 jof-06-00145-f001:**
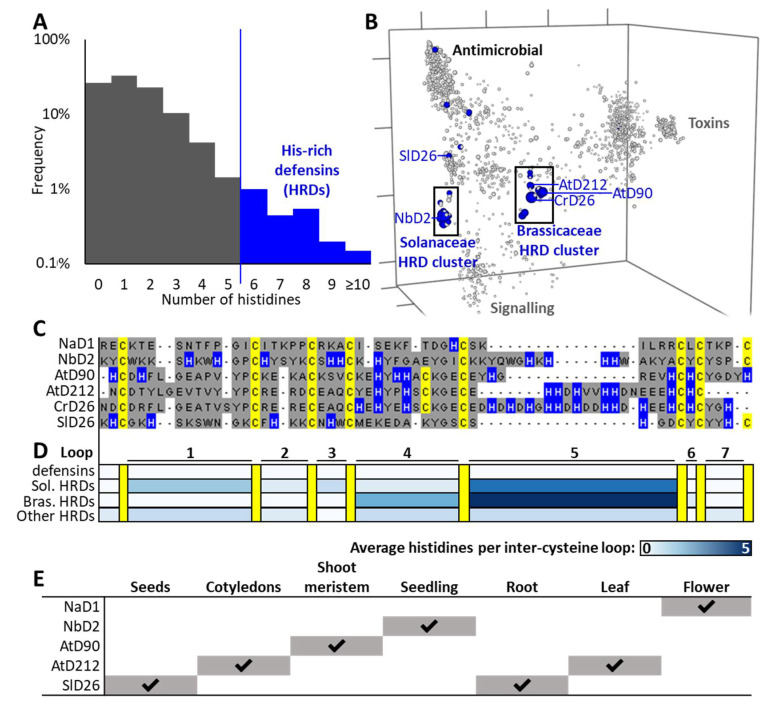
Histidine-rich defensins (HRDs) are a rare subset of the plant defensins. (**A**) Histidine-frequency in plant defensins (log scale). Grey shading represents the bulk of defensin sequences. Blue represents sequences identified as histidine rich. (**B**) Sequence space of 1901 defensins with HRDs (≥6 His) highlighted in blue. (**C**) Multiple sequence alignment of five example HRDs and a typical antimicrobial defensin (NaD1) for comparison. Cysteines are drawn in yellow, histidines in blue, all other residues in grey. Inter-cysteine loops are numbered below as per NaD1. (**D**) Average histidine number per loop for all defensins, HRDs in the Solanaceae cluster, HRDs in the Brassicaceae cluster, and HRDs that sit outside of those clusters. For alignment overviews see Appendix A. (**E**) Tissues where gene expression has been reported in the literature [20,30,31,32,33], including HRDs and NaD1. Check marks indicate tissues where expression has been detected. No data were available for CrD26. For more detail on AtD90, AtD212, and SlD26, see Appendix A.

**Figure 2 jof-06-00145-f002:**
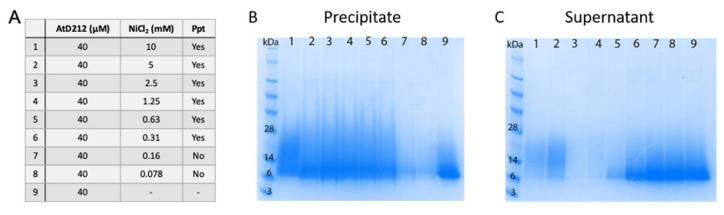
NiCl_2_ induced precipitation of AtD212. (**A**) AtD212 was mixed with a range of concentrations of NiCl_2_ and assessed for precipitation by the eye. Precipitate formed at NiCl_2_ concentrations of 0.31 mM and above. The samples were centrifuged and the precipitate (**B**) and supernatant (**C**) were analyzed by SDS-PAGE. The majority of HRD protein was in the precipitate. At concentrations where there was no precipitation the majority of HRD was in the supernatant. Lane 9 on both gels is 40 µM AtD90 run as a control.

**Figure 3 jof-06-00145-f003:**
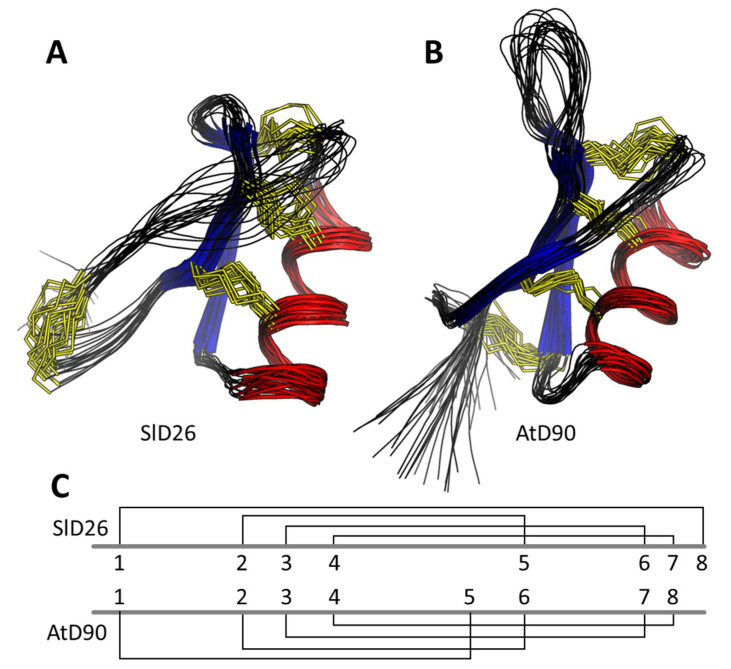
Structural analysis of histidine-rich defensins. Overlay of the 20 lowest-energy states for (**A**) SlD26 and (**B**) AtD90 with disulfides in yellow, beta strands in blue, and alpha helices in red. The differences in disulfide connectivity when comparing SlD26 and AtD90 are shown in panel (**C**). Numbers indicate the order of cysteine residues in the defensin amino acid sequence.

**Figure 4 jof-06-00145-f004:**
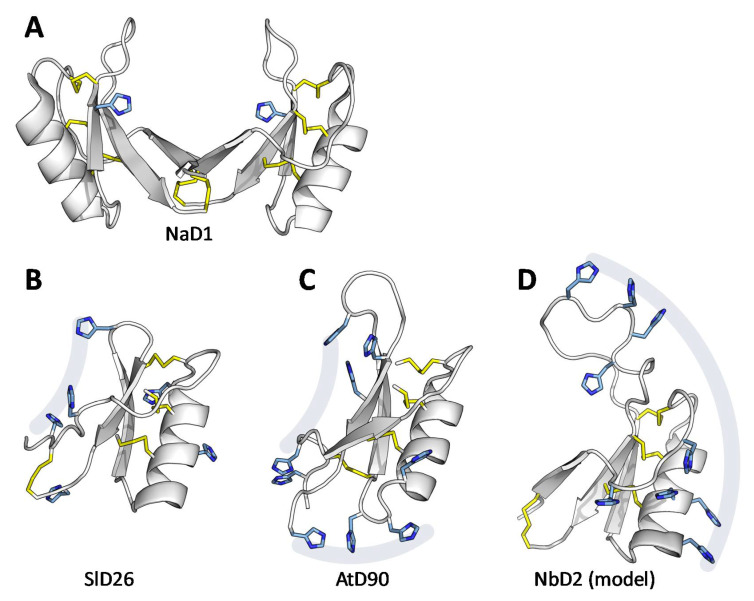
Histidine side-chain orientation. Defensins are shown with disulfides in yellow and histidines in blue (with their nitrogen atoms in dark blue). (**A**) The NaD1 dimer (from PDB:4CQK). (**B**) SlD26, (**C**) AtD90, (**D**) a homology model of NbD2. For flexibility of histidine sidechains see Appendix A).

**Table 1 jof-06-00145-t001:** Sequence and properties of HRDs as expressed and purified.

Protein	Amino Acid Sequence of Defensin Domain *	Total AA	His AA	Mass (Da)	pI
AtD90	AHCDHFLGEAPVYPCKEKACKSVCKEHYHHACKGECEYHGREVHCHCYGDYH	52	9	6038.8	6.5
AtD212	ANCDTYLGEVTVYYPCRERDCEAQCYEHYPHSCKGECEHHDHVVHHDNEEEHCHC	55	9	6548.0	5.1
CrD26	ANDCDRFLGEATVSYPCRERECEAQCHEHYEHSCKGECEDHDHDHGHHDHDDHHDHEEHCHCYGH	65	15	7721.0	5.2
SlD26	AKHCGKHSKSWNGKCFHKKCNHWCMEKEDAKYGSCSHGDCYCYYHC	46	6	5422.2	8.7

* An alanine residue is incorporated into the N-terminus of the expressed defensins as a result of cleavage of the secretion signal in the *P. pastoris* expression system.

**Table 2 jof-06-00145-t002:** Buffer optimization for purification of HRDs.

Protein	Purification Buffer System for IEX					
	Citrate pH 3.0	Acetate pH 4.0	Phosphate pH 6.0	Bis-Tris pH 7.0	Buffer Selected for Purification	Yield (mg/L)
AtD90	-	Good	No Binding	NA	Acetate pH 4	6.3
AtD212	Good	No binding	NA	Precipitation	Citrate pH 3	5.0
CrD26	Good	Partial binding	NA	Precipitation	Citrate pH 3	5.0
SlD26	-	-	Good	-	Phosphate pH 6	16.0

**Table 3 jof-06-00145-t003:** Antifungal activity of HRDs.

Protein	MIC *F. graminearum* (µg/mL)	MIC *C. albicans* (µg/mL)
AtD90	50	25
AtD212	ND	ND
CrD26	ND	ND
SlD26	25	12.5
NaD1	6.25	12.5

ND indicates an MIC greater than the maximum tested concentration of 50 µg/mL. All MIC values were consistent across three independent experiments. An example of the antifungal activity graphs from the microbroth dilution assays is presented in Appendix A.

**Table 4 jof-06-00145-t004:** AtD90 and SlD26 have different metal binding affinities.

Metal Salts	AtD90		SlD26	
	Metal Binding Affinity (*K*_d_ (µM))	Std Error	Metal Binding Affinity (*K*_d_ (µM))	Std Error
NiCl_2_	495	4.3	44	3.9
NaCl	no binding	-	no binding	-
ZnCl_2_	293	11.2	442	5.9
MnCl_2_	230	3.5	no binding	-
MgCl_2_	no binding	-	no binding	-

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
