# Peer review of "Histidine-Rich Defensins from the Solanaceae and Brasicaceae Are Antifungal and Metal Binding Proteins"

_jof, 2020, doi:10.3390/jof6030145_

Round 1
Reviewer 1 Report
Dear Authors
Please consider the below references
Park, C.B. Park, S.S. Hong, H.S. Lee, S.Y. Lee, S.C. Kim, Characterization and cDNA
cloning of two glycine- and histidine-rich antimicrobial peptides from the roots of
shepherd's purse, Capsella bursa-pastoris, Plant Mol. Biol. 44 (2000) 187–197,
http://dx.doi.org/10.1023/A:1006431320677.
Amanda Mangeon, Ricardo Magrani Junqueira & Gilberto Sachetto-Martins
(2010) Functional diversity of the plant glycine-rich proteins superfamily, Plant Signaling &
Behavior, 5:2, 99-104, DOI: 10.4161/psb.5.2.10336
Shaq, N.; Bilal, M.; Iqbal, H.M. Medicinal Potentialities of Plant Defensins: A Review with Applied Perspectives. Medicines 2019, 6, 29.
a pvalue many lower than 0.05 would be preferable.
figures S1, S2 S3 and S4 do not have the captions. Figure S2 shows to be incomprehensible and the quality of the figure is very bad for a careful evaluation
Author Response
We have cited the Ishaq et al paper on line 44 and the Park et al paper on line 65
A p-value of 0.05 is standard
We apologise for the size of the supplemental figures. The settings were wrong when they were exported and compressed. We have now provided the supplementary figures with expanded captions as a pdf to ensure that they are of the appropriate resolution.
Reviewer 2 Report
The manuscript entitled “Histidine-rich defensins from the Solanaceae and Brasicaceae are antifungal and metal binding proteins” by Bleackley et al., demonstrated protein profiling in somatic embryogenesis of oil palm. This manuscript describes a group of plant defensins with enriched histidine residues could be classified as histidine rich defensins (HRDs) based on the authors’ analysis of initial identification and principal component analysis. In addition, the authors synthesize the genes according to six HRDs genes NbD2, NbD3, AtD212, CrD26, and SlD26, and further determined antifungal activity, metal induced precipitation, Microscale thermophoresis, and solution structure of AtD90. Overall, the manuscript is well written and tested those HRDs with structural and functional analysis. However, some points should be identified.
- Although the authors claimed that the selected HRDs seems to express in those tissues in Figure 1E by transcriptomic analysis, how can we know those HRD proteins are actually existed in those tissues? In addition, those HRDs with enriched histidine residues expressed in various tissues, what are the possible explanations for the different distribution?
- The supplemental figures are too small to see. Furthermore, the sequences and labels of supplemental figures should be revised. For example, Figure S2 firstly appeared in line 209, and Supplemental Figure 1 appeared in line 238.
- In the section of antifungal activity of HRDs, the authors just provide the treated concentrations for human fungal pathogen and the cereal pathogen. I am wondering the authors should provide the raw data to assessment in microbroth dilution assays either in supplemental data.
- In the section of structure analysis, the authors determine the solution structures of AtD90, and SLD26. Those two structures should be uploaded to PDB for public purposes. The PDB id also should be included in the manuscript.
- In Figure 2, NiCl2 induced precipitation of AtD212. There are soluble and precipitate forms of AtD212 in Line 9. It means AtD212 would also precipitate in normal condition. In addition, the lower metal concentrations in Line7 and Line8, the precipitate of AtD212 is almost disappeared. The authors should provide explanation for this phenomenon.
The following comments and suggestions should be considered:
- Page 10 Lines 310: remove “)” from the sentence “For flexibility of histidine sidechains see Figure S4).”
- No supplemental tables are provided.
Author Response
- We have now included some text at line 203 acknowledging that we have not examined the proteins in planta. It is likely that this mRNA is translated into protein, but this have not been shown conclusively. The only way to do this is by use of specific antibodies in immunocytochemistry as identifying specific defensins in plant extracts is notorious difficult due to their stability and the presence of multiple defensins in one plant species. These antibodies do not exist and nothing was known about these proteins before we started this study. That is why we decided to produce them recombinantly to learn more about their biochemistry and potential functions
2. We apologise for the quality of the supplemental figures. We have now supplied them as a pdf so we can be confident that they are legible. They have also been renumbered according to the order of appearance.
3. We have included antifungal activity graphs as Figure S4
4. The PDB IDs have been included at line 307
5. Lane 9 in both gels is the control where AtD212 has been loaded directly onto the gel. This is stated in the figure legend at line 276.
The bracket has been removed from the legend of figure 4
Supplementary tables have been included in the pdf with the figures
Round 2
Reviewer 1 Report
Dear Authors
1 Change Brassica to Brassicaceae.
2 Figure 4 is seen twice. Delete the second one.
3 Describe plant material sampling methods and the vegetal sample conservation procedures.
4 Report the order and the execution in detail of the proteomic procedure analysis.
5 Report the order and the execution in detail of the ionomic procedure analysis.
6 Recalculate the significance values by considering the pvalue as in S1 figures
S1, S2 S3 and S4 now have the captions.
Figure S2 is incomprehensible.
Figure S2 quality is very bad .
Then, do figure S4 again
It is reported that For flexibility of histidine sidechains see Figure S4S5).
Figure S5 is not present in the manuscript. Add Figure S5
Bibliografy
Ishaq, N.; Bilal, M.; Iqbal, H.M. Medicinal Potentialities of Plant Defensins: A Review with Applied Perspectives. Medicines 2019, 6, 29. Adjust